# Token-Supervised Value Models for Enhancing Mathematical Problem-Solving Capabilities of Large Language Models

**Jung Hyun Lee**[*,1]**, June Yong Yang**[*,1,2]**, Byeongho Heo**[3]**, Dongyoon Han**[3]**,**
**Kyungsu Kim**[4]**, Eunho Yang**[2,5]**, Kang Min Yoo**[1]
[1] NAVER Cloud   [2] KAIST AI   [3] NAVER AI Lab   [4] SNU   [5] AITRICS
{onliwad101,laoconeth}@gmail.com
{bh.heo,dongyoon.han,kangmin.yoo}@navercorp.com
kyskim@snu.ac.kr,eunhoy@kaist.ac.kr

## Abstract

With the rapid advancement of test-time compute search strategies to improve the mathematical problem-solving capabilities of large language models (LLMs), the need for building robust verifiers has become increasingly important. However, all these inference strategies rely on existing verifiers originally designed for Best-of-N search, which makes them sub-optimal for tree search techniques at test time. During tree search, existing verifiers can only offer indirect and implicit assessments of partial solutions or under-value prospective intermediate steps, thus resulting in the premature pruning of promising intermediate steps. To overcome these limitations, we propose token-supervised value models (TVMs) – a new class of verifiers that assign each token a probability that reflects the likelihood of reaching the correct final answer. This new token-level supervision enables TVMs to directly and explicitly evaluate partial solutions, effectively distinguishing between promising and incorrect intermediate steps during tree search at test time. Experimental results demonstrate that combining tree-search-based inference strategies with TVMs significantly improves the accuracy of LLMs in mathematical problem-solving tasks, surpassing the performance of existing verifiers.

## 1 Introduction

Although recent large language models (LLMs) (Jiang et al., 2023; Dubey et al., 2024) have showcased extensive capabilities across various domains, they still face challenges with complex multi-step reasoning tasks such as mathematical problem-solving. Considering that existing reasoning problems can often be solved by drawing inferences from pre-trained knowledge (Snell et al., 2024), recent studies have focused on inference techniques that invest additional computational effort at test time to better elicit the appropriate knowledge from these models. The simplest and most conventional inference strategy to scale up test-time compute is Best-of-N search (Lightman et al., 2023), which selects one of the $N$ generated solutions based on a specific criterion. For solving math word problems, since no automated tools exist to verify the exact correctness of a candidate solution at test time (as the ground truth answer is unavailable by definition), researchers have introduced the use of neural *verifiers* trained to assess the correctness of the candidate solution in Best-of-N search.

Existing verifiers for Best-of-N search in mathematical problem-solving tasks can be categorized into two types: outcome-supervised reward models (ORMs) and process-supervised reward models (PRMs). ORMs (Cobbe et al., 2021; Uesato et al., 2022) are trained to assess the correctness of a solution by labeling every token in a solution as either correct or incorrect based solely on whether the final answer in the solution is correct. In contrast, PRMs (Uesato et al., 2022; Lightman et al., 2023; Wang et al., 2024b;c; Chen et al., 2024; Luo et al., 2024) are trained with step-level labels to assess the correctness of each intermediate solution step. Thanks to their finer-grained assessment, PRMs are better equipped to diagnose errors even when only a few intermediate steps are incor-

---

[*]Equal contribution.

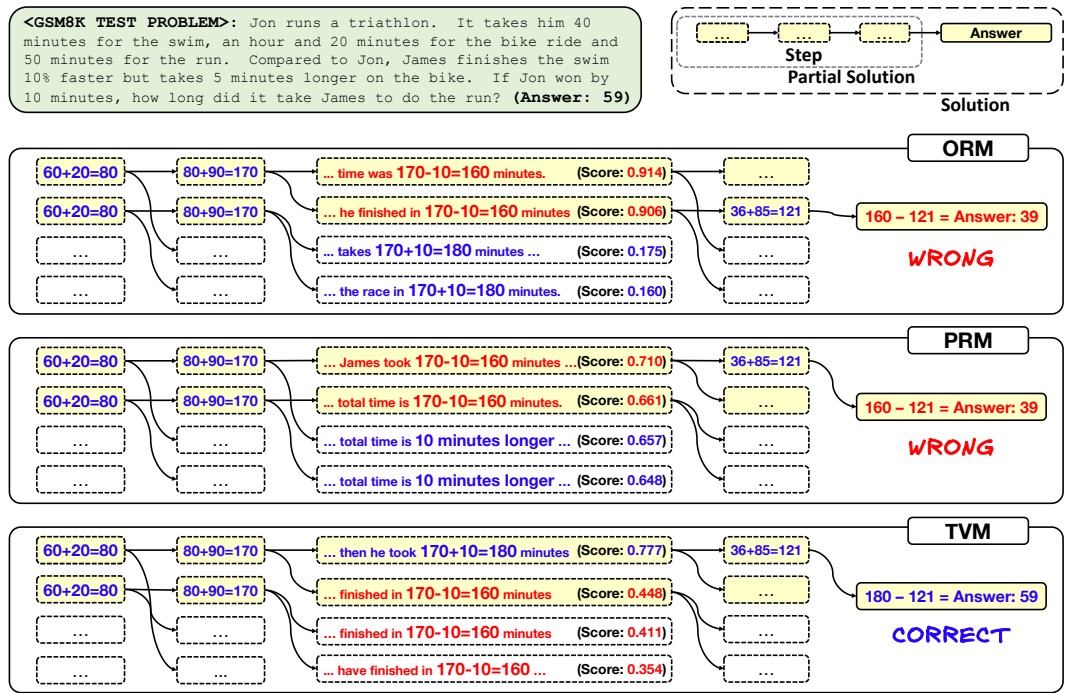

Figure 1: **Illustration of ORM's, PRM's, and TVM's (ours) intermediate steps and their corresponding scores under Step-by-Step Beam Search for a test problem in GSM8K.** The ORM incorrectly predicts wrong intermediate steps (colored red) with excessively high scores (0.914 or 0.906) while assigning very low scores (0.175 or 0.160) to correct steps (colored blue). Although the PRM predicts correct steps with higher scores (0.657 or 0.648) than the ORM, it still assigns comparable scores to correct steps with respect to scores of wrong steps (0.710 or 0.661), causing the premature pruning of promising steps. Yet, the TVM accurately predicts a correct step with a high score (0.777) and wrong steps with relatively low scores (0.448, 0.411, 0.354), thus improving the performance of tree-search-based inference algorithms over both ORM and PRM.

rect, thereby resulting in *a lower false positive error* (Lightman et al., 2023). As a result, PRMs are generally considered more robust and preferable as verifiers than ORMs when using Best-of-N search. Nonetheless, the accuracy of Best-of-N search typically plateaus once $N$ exceeds a few hundred (Brown et al., 2024).

To more effectively utilize additional test-time computation to enhance the mathematical problem-solving capabilities of LLMs, tree search algorithms such as Step-by-Step Beam Search and Monte Carlo Tree Search have been actively explored as alternatives (Yu et al., 2024; Feng et al., 2024; Chen et al., 2024; Wu et al., 2024; Snell et al., 2024) to the Best-of-N search. While Best-of-N search only permits neural verifiers to score solutions after they are fully generated, tree-search-based inference methods enable verifiers to intervene during the solution generation process. This results in improved performance compared to Best-of-N search while utilizing less inference-time computation. Yet, all test-time search strategies rely on existing verifiers (i.e., ORMs and PRMs), which were initially developed for Best-of-N search. As a result, it remains unclear whether both ORMs and PRMs are suited for tree-search-based inference techniques. For instance, when implementing tree-search-based inference strategies, Yu et al. (2024) argued that ORMs are more suitable than PRMs while Wu et al. (2024); Snell et al. (2024) favored PRMs instead of ORMs. In this paper, we will elucidate our assertion that both ORMs and PRMs have drawbacks in utilizing additional computation at inference time with tree search algorithms.

One key property to maximize the accuracy of LLMs in solving math word problems through tree search at inference time is that poor intermediate solution steps must be preemptively filtered out, while promising ones should be preserved and further explored. In turn, the verifier needs to be trained to predict which partial solutions are on the right path toward a correct final answer. However, both ORMs and PRMs possess limitations in achieving this. Specifically, given that ORMs are trained with every token labeled as either correct or incorrect based merely on whether the final answer is correct, they can only infer the potential correctness of a partial solution implicitly and

indirectly. Additionally, since PRMs are trained to determine an entire intermediate step as incorrect even if only the final few tokens are erroneous, they are prone to under-valuing prospective intermediate steps by assigning scores comparable to those of incorrect steps as shown in Figure 1.

In this work, we reveal that PRMs lacking in intra-step supervision systematically exhibit *a high false negative error* in evaluating the correctness of intermediate steps. This phenomenon results in the under-valuing and premature pruning of promising intermediate steps during tree search at test time (see Figure 1), negatively impacting task accuracy. To equip a verifier with a more direct and explicit ability to evaluate partial solutions while minimizing false negative errors, we propose token-supervised value models (TVMs) – a new class of verifiers trained by supervising each token in a solution with the probability of reaching the correct final answer. As TVMs are designed to directly and explicitly predict the potential correctness of a partial solution, they are more effective at evaluating whether a partial solution is on a promising path toward the correct answer than ORMs. In addition, unlike PRMs, TVMs benefit from token-level supervision with distinct correctness probability scores. When labeling tokens in an incorrect intermediate step, only the last few tokens can be labeled as zero (i.e., incorrect) while the rest are assigned with positive values. This enables TVMs to separate promising intermediate steps from incorrect ones (see Figure 1) and thus attain *a lower false negative error* than PRMs, while preserving a false positive error comparable to that of PRMs. Therefore, TVMs demonstrate improved performance in tree-search-based inference strategies over both ORMs and PRMs.

Our contribution is threefold:

- To the best of our knowledge, we are the first to disclose that PRMs produce a high false negative error, which we demonstrate to be detrimental to the performance of tree-search-based inference algorithms due to premature pruning of promising intermediate steps.

- We propose the Token-supervised Value Model (TVM) – a new type of verifiers that are trained to directly and explicitly estimate the likelihood of reaching the correct final answer for each token in a solution. The TVM achieves a lower false negative error than the PRM, while maintaining a false positive error comparable to that of the PRM, which makes the TVM particularly suitable for tree-search-based inference methods (see Figure 1).

- We provide a theoretical insight that the value of each token is equivalent to the probability of reaching the correct final answer given until that token, which leads us to name the token-supervised *value* model, not a reward model.

## 2 TEST-TIME STRATEGIES FOR MATHEMATICAL PROBLEM-SOLVING

This section briefly reviews test-time strategies to enhance the mathematical problem-solving capabilities of large language models (LLMs) by leveraging additional compute at inference time. Here, we discuss two main test-time search strategies: (i) Best-of-N Search and (ii) Tree Search.

**Best-of-N Search.** The most basic approach to test-time compute utilization is to sample $N$ solutions in parallel and choose the one most likely to be correct, which is known as Best-of-N search. For tasks such as code generation and neural theorem proving, each of the $N$ candidate solutions can be automatically identified as either correct or incorrect using unit tests or proof assistants (e.g., Lean 4 (Moura & Ullrich, 2021)), leading to improved pass rates as $N$ increases (Brown et al., 2024). However, for math word problems, determining the correctness of a solution cannot be automated without knowing the ground truth answer at inference time. To address this, a neural verifier is employed to assess the correctness of the $N$ sampled solutions. In this framework, it is critical to control the false positive error of the verifier, as it has to identify the single most probable solution and discard the rest.

**Tree Search.** An advanced method for test-time compute utilization is to alter the reasoning trajectory of an LLM by allowing the verifier to intervene in the intermediate steps of the reasoning process, which can prevent errors in earlier steps from propagating to subsequent steps. To this end, researchers have actively explored Tree Search as a more effective alternative to Best-of-N search. One easy-to-implement and well-studied tree-search-based inference strategy is Step-by-Step Beam Search (Yu et al., 2024; Chen et al., 2024; Snell et al., 2024). It operates as follows: (i) The LLM

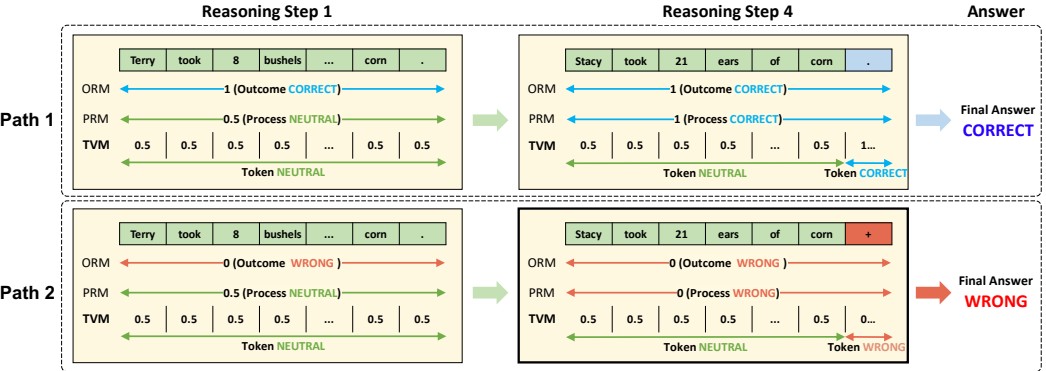

Figure 2: **Illustrative comparison of token-level supervision (TVM; ours) with outcome supervision (ORM) and process supervision (PRM).** We provide two examples for each correct and wrong reasoning path. Outcome supervision employs homogeneous labels judged by the correctness of an entire reasoning path, uniformly labeling all tokens in every reasoning step as either correct or wrong. While process supervision uses a percentage score (i.e., $0.5$) before the reasoning step 4, it assigns the reasoning step 4 to zero due to the incorrectness of only the final few tokens (see the bold box), which causes *a high false negative error*. On the other hand, token-level supervision (ours) allows for labeling only the last few tokens as zero while giving positive values to the remaining tokens of the reasoning step 4, thus achieving *a lower false negative error* than process supervision and distinguishing promising intermediate steps from incorrect ones as demonstrated in Figure 1.

generates $K$ initial reasoning steps in parallel. (ii) The verifier scores the $K$ steps, pruning low-scoring ones and retaining only $b$ steps in the beam. (iii) For each of these $b$ candidates, the next $K/b$ steps are generated in parallel, leading to $b \times K/b = K$ new subsequent steps. (iv) This process is repeated until a stopping criterion is met. A variation of this technique, known as Reward Balanced Search (REBASE), introduced by Wu et al. (2024), balances the pruning and expansion width of the $K$ steps at each depth. Additionally, Monte Carlo Tree Search and its variants have been explored, but multiple studies (Yu et al., 2024; Chen et al., 2024; Snell et al., 2024; Wu et al., 2024) indicate that they generally underperform compared to Step-by-Step Beam Search and REBASE.

## 3 PITFALLS OF OUTCOME AND PROCESS SUPERVISION

In Section 3.1, we outline the limitations of outcome-supervised reward models (ORMs) and process-supervised reward models (PRMs) for tree search strategies at test time by using illustrative examples presented in Figure 1. Section 3.2 provides a brief overview of the preliminary setup for training a neural verifier to enhance the mathematical problem-solving capabilities of large language models (LLMs). In Sections 3.3 and 3.4, we then analyze the issues arising from outcome supervision and process supervision, respectively.

### 3.1 PROBLEM STATEMENT

To maximize LLMs' problem-solving accuracy with tree search at test time, it is crucial to effectively prune poor intermediate steps while exploiting prospective ones. This requires a verifier that can accurately predict which partial solutions are on the right track toward the correct final answer. However, we hypothesize that existing supervision approaches (i.e., outcome and process supervision) have their own inherent limitation when implementing tree search strategies at test time.

Figure 1 illustrates that ORMs assign excessively high scores (e.g., $0.914$ or $0.906$) to incorrect intermediate solution steps while giving totally low scores (e.g., $0.175$ or $0.160$) to correct ones. Although PRMs score correct intermediate steps (e.g., $0.657$ or $0.648$) higher than ORMs, PRMs still assign comparable scores to correct steps with respect to scores of incorrect steps (e.g., $0.710$ or $0.661$), leading to the premature pruning of promising steps. In Sections 3.3 and 3.4, we further validate our hypothesis by demonstrating that ORMs generate less accurate value estimates, while PRMs under-value promising steps due to high false negative errors.

Table 1: Root mean squared error (RMSE) between true value and a verifier's value estimation at the token level on the test set of GSM8K for Mistral 7B and Mistral 7B MetaMath, where a verifier is either ORM or TVM. We approximate true value of a token as Eq. 6 by sampling 256 reasoning paths per test problem, because it is intractable to obtain the ground truth of value due to the infeasibility of calculating the expected returns analytically for all possible paths.

| Method | Mistral 7B | Mistral 7B MetaMath |
|--------|-----------|---------------------|
| ORM | 0.2813 | 0.2471 |
| TVM | **0.2575** | **0.2406** |

## 3.2 Training a Neural Verifier for Mathematical Problem-Solving

Since LLMs are autoregressive models based on next-token prediction, they are incapable of retracting or modifying previously generated outputs. For mathematical problem-solving, reward models can be employed as verifiers to evaluate the correctness of the generated outputs. A verifier is trained via supervised learning on a dataset obtained by sampling multiple solutions per training problem $q_{tr}$ using the LLM. Given a training math word problem $q_{tr}$ as an input, the LLM first generates $N_{tr}$ solutions (or reasoning paths), where the $n$-th reasoning path consists of reasoning steps $\{s_{n,j}\}_{j=1}^{S_n}$ and a final answer $a_n$ for $n = 1, \cdots, N_{tr}$. In token-level notation, the $n$-th reasoning path can also be expressed as a sequence of tokens, denoted by $\{t_{n,k}\}_{k=1}^{T_n}$. The final answer $a$ is *correct* if it is equal to the ground truth answer $\hat{a}$, and *incorrect* otherwise. To train a verifier, supervision is traditionally given in two ways with respect to its granularity: (i) outcome supervision in ORMs (Cobbe et al., 2021; Uesato et al., 2022) and (ii) process supervision in PRMs (Uesato et al., 2022; Lightman et al., 2023; Wang et al., 2024b;c; Chen et al., 2024; Luo et al., 2024).

## 3.3 Outcome Supervision

An ORM (Cobbe et al., 2021; Uesato et al., 2022), $f_{ORM}$ is a verifier trained to model the outcome reward function $r_o(\cdot)$, which is the correctness of a final answer:

$$r_o(a) = \begin{cases} 1 \text{ if } a = \hat{a} \\ 0 \text{ if } a \neq \hat{a}. \end{cases} \tag{1}$$

To train $f_{ORM}$, *outcome supervision* is employed. Given $N_{tr}$ reasoning paths generated for a training problem $q_{tr}$, as described in Figure 2, outcome supervision labels every token in each reasoning path as correct if its final answer is correct, which is precisely the outcome reward (Eq. 1). In turn, the ORM loss for a $\mathcal{L}_{ORM}$ is defined as:

$$\mathcal{L}_{ORM} = \sum_{n}^{N_{tr}} \sum_{k}^{T_n} \ell\left(r_o(a_n), f_{ORM}(q_{tr}, t_{n,1}, t_{n,2}, \cdots, t_{n,k})\right), \tag{2}$$

where the mean squared error is typically used as the loss function $\ell(\cdot)$. Note Cobbe et al. (2021) demonstrated that a token-level verifier trained to judge the correctness for every token in a solution improves over a solution-level verifier trained to determine the correctness only for the final token.

Albeit designed as a reward model for Best-of-N search, ORMs can be alternatively described as modeling *the cumulative reward* for each token, where all intermediate rewards are zero (i.e., $r(t_{n,k}) = 0$ for every $n$ and $k$) and the discount factor $\gamma$ is set to 1 (Yu et al., 2024). The cumulative reward following an intermediate token $t_{n,k}$, $R(t_{n,k}) = \sum_{l=1}^{\infty} \gamma^{l-1} r(t_{n,k+l})$ is calculated as

$$R(t_{n,k}) = r(t_{n,k+1}) + \cdots + r(t_{n,T_n}) + r_o(a_n) = \begin{cases} 0 + \cdots + 0 + 1 = 1 & \text{if } a_n = \hat{a} \\ 0 + \cdots + 0 + 0 = 0 & \text{if } a_n \neq \hat{a}, \end{cases} \tag{3}$$

which is equivalent to $r_o(a_n)$ in Eq. 1. This implies that an intermediate reasoning path is labeled as correct if the final answer is correct, and vice versa. In this sense, Yu et al. (2024) showed that ORMs can indirectly and implicitly learn the potential correctness of an intermediate reasoning path.

However, such implicit supervision through homogeneous token labeling renders ORMs to still fall short in precisely evaluating whether an intermediate reasoning path is on a promising track towards the correct final answer. This can be corroborated by not just Figure 1 but also Table 1, which shows that the ORM yields less accurate value estimates compared to the TVM.

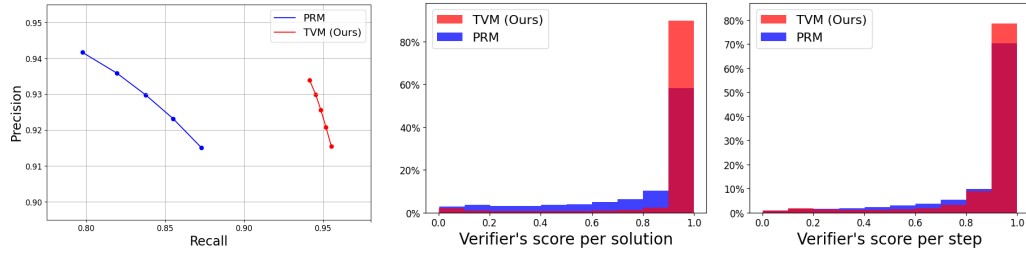

(a) Precision-Recall Curve    (b) Verifier's Scores Histogram for Correct Sampled Solutions/Steps

Figure 3: **Comparative analysis of PRM and TVM (ours)** by sampling 256 solutions per test problem in GSM8K and gathering all sampled solutions (i.e., $256 \times 1319 = 337664$ solutions). (a) The precision-recall curve of PRM and TVM by adjusting the classification threshold between 0.4 and 0.6 in increments of 0.05. (b) Histogram of a verifier's scores for correct sampled solutions (left) and their corresponding steps (right).

### 3.4 PROCESS SUPERVISION

To outperform outcome supervision for Best-of-N search, *process supervision* enables step-wise assessments of a reasoning path through explicit training on the correctness of each individual reasoning step, which is finer-grained supervision than outcome supervision. The correctness of each step is either labeled via human annotation (Uesato et al., 2022; Lightman et al., 2023) or automation (Wang et al., 2024b;c; Chen et al., 2024; Luo et al., 2024). Since acquiring human annotations is labor-intensive and costly, we mainly focus on process supervision without human annotations.

Following Wang et al. (2024b), $s_{n,j}$ is annotated by sampling a fixed number of reasoning paths conditioned on a sequence of intermediate reasoning steps $s_{n,1}, \cdots, s_{n,j}$. If all of the sampled reasoning paths reach wrong final answers, $s_{n,j}$ is labeled as incorrect with the process reward $r_p(s_{n,j}) = 0$. Otherwise, $s_{n,j}$ can be labeled as either $r_p(s_{n,j}) = 1$ (i.e., correct) or the probability of the sampled reasoning paths reaching the correct final answer. Using the per-step labels obtained through automation, a PRM is trained to provide a step-level assessment by minimizing the following loss:

$$\mathcal{L}_{PRM} = \sum_{n}^{N_{tr}} \sum_{j}^{S_n} \ell\left(r_p(s_{n,j}), f_{PRM}(q_{tr}, s_{n,1}, s_{n,2}, \cdots, s_{n,j})\right), \tag{4}$$

where $\ell$ denotes the binary cross entropy loss.

This form of step-level supervision improves the identification of errors even when only a few intermediate steps are erroneous, leading PRMs to have low false positive errors (i.e., high precision), as discussed in Lightman et al. (2023). However, we observe that PRMs without human annotations suffer from high false negative errors (i.e., low recall), because process supervision labels an entire intermediate step as incorrect even if only the final few tokens are wrong (see Figure 2). As a result, as shown in Figure 3(a), PRMs without human annotations exhibit significantly lower recall compared to our proposed verifier, the TVM. Hereafter, we refer to PRMs without human annotations simply as PRMs to keep the expression concise.

To further investigate the PRM's recall, we compare the scores assigned by the PRM and TVM to correct sampled solutions (the left side of Figure 3(b)) and their corresponding steps (the right side of Figure 3(b)). Due to the PRM's significantly lower recall, its overall solution scores are naturally lower than those of the TVM. Surprisingly, the PRM also assigns lower scores to individual correct steps compared to the TVM, resulting in a smaller proportion of steps being scored close to one. This supports that the PRM tends to under-value promising intermediate solution steps, thus resulting in premature pruning during tree search at inference time.

## 4 METHOD

This section introduces our proposed verifier, the Token-supervised Value Model (TVM), which is based on a new token-level supervision approach to directly and explicitly estimate the probability of reaching the final answer for each token along a reasoning path. We first outline how to empirically compute per-token correctness probability scores from the $N_{tr}$ generated reasoning paths for token-level supervision. Then, we provide a theoretical insight into our proposed verifier as a *value* model.

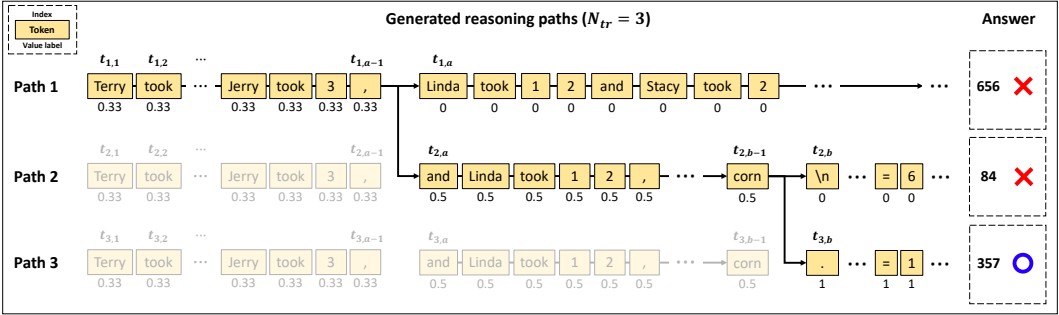

Figure 4: **Illustration of token-level supervision with correctness probability scores using Eq. 6.** For a single training problem $q_{tr}$, $N_{tr}$ reasoning-answer pairs are sampled using an LLM. Here, let $N_{tr} = 3$ for convenience. (1) All three sentences begin with the same tokens $\{t_{1,k}\}_{k=1}^{a-1}$, and only one of them reaches the correct final answer (357). Accordingly, every token of $\{t_{1,k}\}_{k=1}^{a-1}$ is labeled as $1/3 = 0.33$. (2) At the $a$-th position, however, only one sentence starts with $t_{1,a}$, which reaches an incorrect final answer (656). Thus, all tokens after $t_{1,a}$ are labeled as $0/1 = 0$. (3) The remaining two sentences continue with the same tokens $\{t_{2,k}\}_{k=a}^{b-1}$, only one of which is correct. Hence, every token of $\{t_{2,k}\}_{k=a}^{b-1}$ is labeled as $1/2 = 0.5$. (4) Finally, at the $b$-th position, which one is correct is pre-determined. As a result, all tokens after $t_{2,b}$ are labeled as $0/1 = 0$, whereas all tokens after $t_{3,b}$ as $1/1 = 1$.

## 4.1 TOKEN-LEVEL SUPERVISION WITH CORRECTNESS PROBABILITY SCORES

To supervise each token in a reasoning path according to its potential in deducing the correct final answer, we label each token as the probability of reaching the correct final answer conditioned on until that token. To be more concrete, we label an intermediate token $t_{n,k}$ as

$$\mathbb{P}(\text{the final answer will be } \hat{a}|q_{tr}, t_{n,1}, \cdots, t_{n,k}) = \frac{\mathbb{P}(t_{n,1}, \cdots, t_{n,k} \cap \text{the final answer will be } \hat{a}|q_{tr})}{\mathbb{P}(t_{n,1}, \cdots, t_{n,k}|q_{tr})} \quad (5)$$

for $k = 1, \cdots, T_n$ and $n = 1, \cdots, N_{tr}$. Let $\{t_{n,1}, \cdots, t_{n,k}\}$ be $\{t_{n,k'}\}_{k'=1}^{k}$ interchangeably. In practice, from $N_{tr}$ generated reasoning paths, Eq. 5 can be empirically computed as the ratio of correct reasoning paths starting from $\{t_{n,k'}\}_{k'=1}^{k}$ among $N_{tr}$ to total reasoning paths starting from $\{t_{n,k'}\}_{k'=1}^{k}$ among $N_{tr}$, respectively. Hence, the label of each token $t_{n,k}$ can be assigned with

$$\mathbb{P}(\text{the final answer will be } \hat{a}|q_{tr}, \{t_{n,k'}\}_{k'=1}^{k}) = \frac{\sum_{n'=1}^{N_{tr}} \mathbb{I}(\{t_{n,k'}\}_{k'=1}^{k} = \{t_{n',k'}\}_{k'=1}^{k} \cap a_{n'} = \hat{a})/N_{tr}}{\sum_{n'=1}^{N_{tr}} \mathbb{I}(\{t_{n,k'}\}_{k'=1}^{k} = \{t_{n',k'}\}_{k'=1}^{k})/N_{tr}}, \quad (6)$$

where $\mathbb{I}(\cdot)$ is the indicator function and $N_{tr}$ cancels out in the right hand side. Finally, the token-supervised value model (TVM), $f_{TVM}$ is trained by minimizing the following loss using Eq. 6:

$$\mathcal{L}_{TVM} = \sum_{n}^{N_{tr}} \sum_{k}^{T_n} \ell\left(\mathbb{P}(\text{the final answer will be } \hat{a}|q_{tr}, t_{n,1}, \cdots, t_{n,k}), f_{TVM}(q_{tr}, t_{n,1}, \cdots, t_{n,k})\right), \quad (7)$$

where $\ell$ is the mean squared error.

Thanks to this new token-level supervision with Eq. 5, the TVM is trained to directly and explicitly evaluate whether an intermediate reasoning path (i.e., $\{t_{n,k'}\}_{k'=1}^{k}$) is on a promising track toward the correct final answer, thereby producing more accurate value estimates than the ORM as shown in Table 1. Not only that, as illustrated in Figure 3(a), the TVM can also achieve a lower false negative error than the PRM with a reduction ranging from 8 to 14%p, while maintaining a comparable false positive error (within 1%p of the PRM). The overall procedure and algorithm of calculating Eq. 6 are outlined in Figure 4 and detailed in Appendix B, respectively.

Although Eq. 5 can be computed without sampling multiple roll-outs to train PRMs, Eq. 5 is generally calculated by sampling multiple roll-outs given $q_{tr}$ and $\{t_{n,k'}\}_{k'=1}^{k}$ like Wang et al. (2024b;c); Chen et al. (2024); Luo et al. (2024) do for training PRMs. It is worth noticing that the multiple roll-out process per token requires $O(\sum_{n=1}^{N_{tr}} \sum_{k=1}^{T_n} T_n - k) = O(\sum_{n=1}^{N_{tr}} T_n^2)$ token generation, as $T_n - k$ tokens need to be generated per roll-out for each $n$ and $k$. However, Eq. 6 can be easily calculated once $N_{tr}$ reasoning paths are sampled, requiring only $O(\sum_{n=1}^{N_{tr}} T_n)$ token generation.

## 4.2 THEORETICAL INSIGHT: PROBABILITY SCORING AS VALUE MODELING

For a token $t_{n,k}$ in an intermediate reasoning path $\{t_{n,1}, \cdots, t_{n,k}\} = \{t_{n,k'}\}_{k'=1}^{k}$, the expected cumulative reward (i.e., value) is written as

$$V(t_{n,k}) = \mathbb{E}\big[\sum_{l=1}^{\infty} \gamma^{l-1} r(t_{n,k+l}) \big| q_{tr}, t_{n,1}, \cdots, t_{n,k}\big], \tag{8}$$

where $r(\cdot)$ and $\gamma$ denote a reward function and the discount factor, respectively. For conventional settings of reinforcement learning with LLMs (Wang et al., 2024a) ($\gamma = 1$ and no intermediate rewards), under the specific outcome reward formulation of Eq. 1, the expected cumulative reward (i.e., value in Eq. 8) can reduce to the probability of reaching the correct final answer conditioned on the question $q_{tr}$ and intermediate reasoning path $\{t_{n,k'}\}_{k'=1}^{k}$, which can be straightforwardly computed from generated reasoning paths (Section 4.1).

**Proposition 4.1.** *Let the reward function $r(t_{n,k})$ be defined as Eq. 1, which includes only the outcome reward and no intermediate reward (i.e., $r(t_{n,k}) = 0$ except the final answer). Then, with the discount factor $\gamma = 1$, the expected cumulative reward (i.e., value in Eq. 8) is equivalent to the probability of reaching the correct final answer conditioned on $q_{tr}$ and $\{t_{n,1}, \cdots, t_{n,k}\} = \{t_{n,k'}\}_{k'=1}^{k}$:*

$$\mathbb{E}\big[\sum_{l=1}^{\infty} \gamma^{l-1} r(t_{n,k+l}) \big| q_{tr}, t_{n,1}, \cdots, t_{n,k}\big] = \mathbb{P}(\text{the final answer will be } \hat{a} | q_{tr}, t_{n,1}, \cdots, t_{n,k}). \tag{9}$$

In light of Proposition 4.1, we coin our proposed verifier as the token-supervised *value* model (TVM), not a reward model. Not only that, given that tree search is fundamentally intended to be guided by *value* rather than *reward*, Proposition 4.1 is important, as Proposition 4.1 guarantees that TVMs allow tree search algorithms to be value-guided.

## 5 EXPERIMENTS

To demonstrate the effectiveness of our proposed verifier, the token-supervised value model (TVM), in solving math word problems, we conduct experiments on the GSM8K (Cobbe et al., 2021) and MATH (Hendrycks et al., 2021) benchmarks. To perform tree search at test time, we use two different tree search methods, Step-by-Step Beam Search and REward BAlanced SEarch (REBASE) (Wu et al., 2024), which generally outperform other strategies such as Monte Carlo Tree Search or its variants, as observed in Yu et al. (2024); Chen et al. (2024); Snell et al. (2024); Wu et al. (2024). Unless otherwise specified, we set $K{=}40$ for Step-by-Step Beam Search and REBASE, and $N{=}256$ for self-consistency (Wang et al., 2023) and Best-of-N search following Wang et al. (2024b). Regardless of which search strategy to use, we choose the single solution ranked highest by a verifier as our final solution.

Our experiments are based on the following LLMs: (1) Mistral 7B (Jiang et al., 2023), Llama 3 8B (AI@Meta, 2024) and (2) those fine-tuned on MetaMATH (Yu et al., 2023). We opt for LLMs under 10B parameters which is a more interesting setting for experimental research considering the observation that using more test-time compute with smaller language models can surpass using less test-time compute with larger language models (Wu et al., 2024; Snell et al., 2024). For all experiments, a verifier is of the same size and architecture as the LLM. In the case of ORMs and TVMs, following Cobbe et al. (2021), a verifier is extended with a scalar head composed of a single gain parameter and a single bias parameter. For PRMs without human annotations, we employ Math-Shepherd (Wang et al., 2024b). As all experimental results in Wang et al. (2024b) are only based on LLMs fine-tuned on MetaMATH, we compare TVM with Math-Shepherd only for Mistral 7B MetaMath and Llama 3 8B MetaMath. For convenience, we call Math-Shepherd PRM hereafter. More experimental details are deferred to Appendix C.

### 5.1 GRADE SCHOOL MATHEMATICS (GSM8K)

Table 2 displays the comparison of TVM with both ORM and PRM under self-consistency, Best-of-N search, Step-by-Step Beam Search, and REBASE on the GSM8K benchmark for Mistral 7B, Mistral 7B MetaMath, Llama 3 8B, and Llama 3 8B MetaMath. For Step-by-Step Beam Search

Table 2: Accuracy of Mistral 7B, Mistral 7B MetaMath, Llama 3 8B, and Llama 3 8B MetaMath on the GSM8K benchmark under self-consistency ($N = 256$), Best-of-N search ($N = 256$), Step-by-Step Beam Search ($K = 40$, $b = 10$), and REBASE ($K = 40$). A verifier shares the same model size and architecture as the LLM. A bold number means the best accuracy under the same test-time search strategy. A boxed number and an underlined number represent the best accuracy and the second best accuracy respectively, irrespective of the choice of a test-time search strategy. Three random trials are conducted to compute the mean accuracy and standard deviation of TVM.

| Search Strategy | Method | Mistral 7B | Mistral 7B MetaMath | Llama 3 8B | Llama 3 8B MetaMath |
|---|---|---|---|---|---|
| | Self-Consistency | 79.23 | 83.90 | 80.97 | 85.44 |
| Best-of-N Search | ORM | 85.52 | 87.41 | 87.79 | 89.77 |
| | PRM | - | 88.55 | - | 90.30 |
| | TVM (Ours) | **88.17** | **89.01** | **88.70** | **90.37** |
| Step-by-Step Beam Search | ORM | 86.73 | 87.79 | 88.10 | 89.69 |
| | PRM | – | 86.66 | – | 88.93 |
| | TVM (Ours) | **87.69**±**0.22** | **88.70**±**0.16** | **89.06**±**0.07** | **90.35**±**0.19** |
| REBASE | ORM | 86.81 | 88.40 | 87.49 | 89.39 |
| | PRM | – | 86.28 | – | 88.70 |
| | TVM (Ours) | **87.97**±**0.16** | **89.21**±**0.14** | **88.60**±**0.09** | **89.84**±**0.21** |

Table 3: Accuracy of Mistral 7B MetaMath, and Llama 3 8B MetaMath on the MATH benchmark under self-consistency ($N = 256$), Best-of-N search ($N = 256$), Step-by-Step Beam Search ($K = 40$, $b = 10$), and REBASE ($K = 40$). A verifier shares the same model size and architecture as the LLM. A bold number means the best accuracy under the same test-time search strategy. A boxed number and an underlined number represent the best accuracy and the second best accuracy respectively, irrespective of the choice of a test-time search strategy. Three random trials are conducted to compute the mean accuracy and standard deviation of TVM.

| Search Strategy | Method | Mistral 7B MetaMath | Llama 3 8B MetaMath |
|---|---|---|---|
| | Self-Consistency | 35.10 | 42.40 |
| Best-of-N Search | ORM | 36.40 | 43.80 |
| | PRM | 37.30 | **44.40** |
| | TVM (Ours) | **37.40** | 43.40 |
| Step-by-Step Beam Search | ORM | 36.80 | 42.40 |
| | PRM | 36.80 | 42.20 |
| | TVM (Ours) | **39.33**±**0.19** | **45.00**±**0.59** |
| REBASE | ORM | 37.20 | 42.20 |
| | PRM | 37.60 | 41.80 |
| | TVM (Ours) | **38.73**±**0.09** | **43.60**±**0.28** |

and REBASE, TVM consistently outperforms other baseline verifiers. Notably, TVM considerably outperforms PRM, which implies that using the PRM negatively affects the performance of tree search at inference time. Surprisingly, TVM also improves over ORMs an PRMs even for Best-of-N search. It is worth noticing that using Step-by-Step Beam Search or REBASE with the TVM can perform closely or even surpass to the level of using Best-of-N search with TVM while spending about $6\times$ less FLOPs and $2\times$ less execution time as indicated in Table 5.

## 5.2 Advanced Mathematics (MATH)

Following Lightman et al. (2023); Wang et al. (2024b), we also use 500 test MATH problems for evaluation, which is the same test dataset of Lightman et al. (2023), incorporating the remaining 4500 test problems into the training dataset of MATH.

In Table 3, the experimental results of TVM are compared with those of ORM and PRM under self-consistency, Best-of-N search, Step-by-Step Beam Search, and REBASE on the MATH benchmark for Mistral 7B MetaMath and Llama 3 8B MetaMath. Although the PRM surpasses the TVM by 1.0%p under Best-of-N search for Llama 3 8B MetaMath, it is noteworthy that using Step-by-Step Beam Search with the TVM outperforms the best accuracy under Best-of-N search for both Mistral

Table 4: FLOPs and execution time of sampling $N_{tr}$ reasoning paths to train TVMs on the GSM8K ($N_{tr} = 100$) and MATH ($N_{tr} = 25$) benchmarks for Mistral 7B MetaMath without and with vLLM.

|                     | GSM8K FLOPs             | GSM8K Time | MATH FLOPs              | MATH Time  |
| ------------------- | ----------------------- | ---------- | ----------------------- | ---------- |
| Sampling w/o vLLM   | $130.4 \times 10^{13}$  | 8.2 hours  | $204.1 \times 10^{13}$  | 20.3 hours |
| Sampling w/ vLLM    | $130.4 \times 10^{13}$  | 4.6 hours  | $204.1 \times 10^{13}$  | 5.7 hours  |

Table 5: FLOPs and execution time of Best-of-N search ($N = 256$) without and with vLLM, Step-by-Step Beam Search ($K = 40$, $b = 10$), and REBASE ($K = 40$) on the GSM8K and MATH benchmarks for Mistral 7B MetaMath.

| Search Strategy              | GSM8K FLOPs            | GSM8K Time | MATH FLOPs             | MATH Time  |
| ---------------------------- | --------------------- | ---------- | ---------------------- | ---------- |
| Best-of-N search w/o vLLM    | $589.3 \times 10^{12}$ | 6.5 hours  | $871.0 \times 10^{12}$  | 22.0 hours |
| Best-of-N search w/ vLLM     | $589.3 \times 10^{12}$ | 2.1 hours  | $871.0 \times 10^{12}$  | 2.4 hours  |
| Step-by-Step Beam Search     | $\mathbf{94.3 \times 10^{12}}$ | **0.9** hours | $\mathbf{142.9 \times 10^{12}}$ | **1.1** hours |
| REBASE                       | $\mathbf{94.3 \times 10^{12}}$ | 1.3 hours  | $\mathbf{142.9 \times 10^{12}}$ | 2.6 hours  |

7B MetaMath and Llama 3 8B MetaMath even with approximately $6\times$ less FLOPs and $2\times$ less execution time as seen in Table 5.

## 5.3 COMPUTE ANALYSIS

**Training Compute Analysis.** To illustrate the computational cost for training TVMs, we estimate floating point operations (FLOPs) and measure the execution time required to sample $N_{tr}$ reasoning paths. These measurements were conducted using $8\times$NVIDIA A100-80GB GPUs for Mistral 7B MetaMath on GSM8K ($N_{tr} = 100$) and MATH ($N_{tr} = 25$), as reported in Table 4. Since sampling $N_{tr}$ reasoning paths has a linear complexity as explained in Section 4.1, the sampling process takes at most less than a day even without LLM serving engines such as vLLM (Kwon et al., 2023). With vLLM, the sampling process can be accelerated by at least a factor of two.

**Inference Compute Analysis.** To compare the inference computation between Best-of-N search, Step-by-Step Beam Search, and REBASE, for Mistral 7B MetaMath, we estimate floating point operations (FLOPs) following Wu et al. (2024); Snell et al. (2024) and measure the execution time when using $8\times$NVIDIA A100-80GB GPUs on GSM8K and MATH in Table 5. Since $N{=}256$ is much larger than $K{=}40$, Best-of-N search consumes much more FLOPs than Step-by-Step Beam Search and REBASE. As Step-by-Step Beam Search uses the same $K{=}40$ as REBASE, the estimated FLOPs are the same for both Step-by-Step Beam Search and REBASE. Nonetheless, Step-by-Step Beam Search spends less execution time than REBASE due to the fact that the $b$ steps are uniformly expanded in parallel with $K/b$ children as delineated in Section 2.

## 6 CONCLUSION

In this work, we reveal inherent limitations in existing neural verifiers for mathematical problem-solving, namely outcome-supervised reward models (ORMs) and process-supervised reward models (PRMs), when applied to tree search algorithms at inference time. This is because both ORMs and PRMs were originally designed for Best-of-N search. Consequently, ORMs can only infer the correctness of partial solutions indirectly and implicitly, while PRMs suffer from high false negative errors, leading them to under-value promising intermediate steps. To overcome these issues, we propose token-supervised value models (TVMs), a new class of verifiers trained with token-level supervision, where each token is assigned a probability reflecting the likelihood of reaching the correct final answer. This new token-level supervision approach enables TVMs to more accurately assess which partial solutions are on a right track than ORMs, and to attain lower false negative errors than PRMs while maintaining comparable false positive errors. As a result, TVMs significantly enhance the performance of tree search algorithms at test time over both ORMs and PRMs.

ACKNOWLEDGMENTS

This work was partly supported by Institute of Information & communications Technology Planning & Evaluation (IITP) grant funded by the Korea government (MSIT) [NO.RS2021-II211343, Artificial Intelligence Graduate School Program (Seoul National University), and No. 2022-0-00984, Development of Artificial Intelligence Technology for Personalized Plug-and-Play Explanation and Verification of Explanation].

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

## A    PROOF OF PROPOSITION 4.1

Let the reward function $r(t_{n,k})$ be defined as Eq. 1, which includes only the outcome reward and no intermediate reward (i.e., $r(t_{n,k}) = 0$ except the final answer). Then, with the discount factor $\gamma = 1$, $\sum_{l=1}^{\infty} \gamma^{l-1} r(t_{n,k+l}) = \sum_{l=1}^{\infty} r(t_{n,k+l})$ becomes either one or zero, depending on whether the resulting final answer will be $\hat{a}$ or not, respectively. As a result, the expected cumulative reward (value in Eq. 8) can be written as

$$\mathbb{E}\Big[\sum_{l=1}^{\infty} \gamma^{l-1} r(t_{n,k+l}) \Big| q_{tr}, t_{n,1}, \cdots, t_{n,k}\Big]$$

$$= \mathbb{E}\Big[\sum_{l=1}^{\infty} r(t_{n,k+l}) \Big| q_{tr}, t_{n,1}, \cdots, t_{n,k}\Big] \quad (\because \gamma = 1)$$

$$= \sum_{r=0}^{1} r * \mathbb{P}\Big(\sum_{l=1}^{\infty} r(t_{n,k+l}) = r \Big| q_{tr}, t_{n,1}, \cdots, t_{n,k}\Big) \quad (\because \sum_{l=1}^{\infty} r(t_{n,k+l}) = 0 \text{ or } 1)$$

$$= \mathbb{P}\Big(\sum_{l=1}^{\infty} r(t_{n,k+l}) = 1 \Big| q_{tr}, t_{n,1}, \cdots, t_{n,k}\Big)$$

$$= \mathbb{P}(\text{the final answer will be } \hat{a} | q_{tr}, t_{n,1}, \cdots, t_{n,k}),$$

because $\sum_{l=1}^{\infty} r(t_{n,k+l}) = 1$ only if the resulting final answer will be $\hat{a}$.

## B   ALGORITHM FOR TOKEN-LEVEL SUPERVISION WITH CORRECTNESS PROBABILITY SCORES

---

**Algorithm 1** Token-level Supervision with Correctness Probability Scores

---

**Require:** For a question $q_{tr}$, $N_{tr}$ reasoning paths, each consisting of $\{t_{n,k}\}_{k=1}^{T_n}$ and a final answer $a_n$, the ground truth answer $\hat{a}$, and the outcome reward function $r_o(a_n)$ in Eq. 1 for $n = 1, \cdots, N_{tr}$.

**Ensure:**

  $H \leftarrow$ dict()

  **for** $n = 1, \cdots, N_{tr}$ **do**

    **for** $k = 1, \cdots, T_n$ **do**

      **if** not $H$.containsKey$[t_{n,1}, \cdots, t_{n,k}]$ **then**

        $H$.insert$([t_{n,1}, \cdots, t_{n,k}], (r_o(a_n), 1))$

      **else**

        $(c, t) \leftarrow H$.get$[t_{n,1}, \cdots, t_{n,k}]$

        $H$.insert$([t_{n,1}, \cdots, t_{n,k}], (c + r_o(a_n), t + 1))$

      **end if**

    **end for**

  **end for**

  **for** $n = 1, \cdots, N_{tr}$ **do**

    **for** $k = 1, \cdots, T_n$ **do**

      $(c, t) \leftarrow H$.get$[t_{n,1}, \cdots, t_{n,k}]$

      // $c$ means the number of correct reasoning paths starting from $t_{n,1}, \cdots, t_{n,k}$

      // $t$ indicates the number of total reasoning paths starting from $t_{n,1}, \cdots, t_{n,k}$

      $V(t_{n,k}) = \frac{c}{t}$                                      $\triangleright$ Eq. 6

    **end for**

  **end for**

---

## C    IMPLEMENTATION DETAILS

In Section 5.1, following Cobbe et al. (2021), an LLM is fine-tuned on the training dataset of GSM8K for two epochs with a batch size of $128$ and a learning rate of $1e$-5. Then, we sample $N_{tr} = 100$ reasoning paths per training problem. In Section 5.2, an LLM fine-tuned on MetaMath generates $N_{tr} = 25$ reasoning paths for each training problem. We generate $N_{tr}$ reasoning paths with a temperature of $0.7$, a top-k of $50$, and a top-p of $1.0$.

We employ the same architecture as Cobbe et al. (2021), a language model extended with a scalar head composed of a single gain parameter and a single bias parameter, to output a score for each token in a reasoning path. In addition, following Cobbe et al. (2021), we use both a language modeling objective and the verification objective, with $20\%$ dropout (Srivastava et al., 2014). We use the AdamW optimizer (Loshchilov & Hutter, 2019) with a linear scheduler to train a verifier. Note that in all experiments, a verifier shares the same model size and architecture as the LLM used to generate the $N_{tr}$ reasoning paths.

Table 6: Learning rate and batch size for training the TVM (ours) when using Mistral 7B, Mistral 7B MetaMath, Llama 3 8B, and Llama 3 8B MetaMath to generate $N_{tr} = 100$ reasoning paths per training problem in GSM8K in Section 5.1.

|  | Mistral-7B | Mistral-7B-MetaMath | Llama3-8B | Llama3-8B-MetaMath |
|---|---|---|---|---|
| Learning rate | 2$e$-6 | 2$e$-6 | 1e-5 | 2$e$-6 |
| Batch size | 512 | 512 | 512 | 512 |

Table 7: Learning rate and batch size for training the TVM (ours) when using Mistral 7B MetaMath and Llama 3 8B MetaMath to generate $N_{tr} = 25$ reasoning paths per training problem in MATH in Section 5.2.

|  | Mistral-7B-MetaMath | Llama3-8B-MetaMath |
|---|---|---|
| Learning rate | 2$e$-6 | 2$e$-6 |
| Batch size | 512 | 512 |

For Best-of-N search, Step-by-Step Beam Search, and REBASE, we use a temperature of $0.7$, a top-k of $50$, and a top-p of $1.0$. The maximum new token length is set to $400$ for GSM8K and $1024$ for MATH, respectively.

# D  DISCUSSION: TOKEN-SUPERVISED SIGNAL IN TVMS FOR THE LATTER PART OF REASONING PATHS

If reasoning paths were sampled with a small $N_{tr}$, the likelihood of reasoning paths overlapping after the first couple of dozen (e.g., 15–20) tokens could be low. In such a case, the token-supervision signal for the latter part of each reasoning path would be labeled uniformly as either 1 or 0, depending on whether the final answer is correct.

However, with a large enough $N_{tr}$ (e.g., $N_{tr} = 100$ for GSM8K as in Cobbe et al. (2021)), we can obtain reasoning paths that overlap significantly beyond the first 15-20 tokens, up to the later steps of reasoning. As demonstrated in Figure 2, reasoning paths can overlap up to the middle of the fourth step - which consists of the first **93** tokens in terms of token count. Below, we provide the complete version of the two overlapping reasoning paths that were illustrated in Figure 2. Bold text indicates the overlapped part between two reasoning paths in Figure 2, which is **93** tokens in total.

---

**Reasoning path 1 in Figure 2**

**Terry took 8 bushels * 14 ears of corn per bushel = 112 ears of corn.**

**Jerry took 3 bushels * 14 ears of corn per bushel = 42 ears of corn.**

**Linda took 12 bushels * 14 ears of corn per bushel = 168 ears of corn.**

**Stacy took 21 ears of corn**.

Altogether, Bob's friends took 112 + 42 + 168 + 21 = 343 ears of corn.

Thus, Bob is left with 50 bushels * 14 ears of corn per bushel – 343

ears of corn = 357 ears of corn.

---

**Reasoning path 2 in Figure 2**

**Terry took 8 bushels * 14 ears of corn per bushel = 112 ears of corn.**

**Jerry took 3 bushels * 14 ears of corn per bushel = 42 ears of corn.**

**Linda took 12 bushels * 14 ears of corn per bushel = 168 ears of corn.**

**Stacy took 21 ears of corn** + 168 ears of corn = 189 ears of corn.

Bob had 50 bushels * 14 ears of corn per bushel = 700 ears of corn to

start with.

So, Bob has 700 ears of corn { 112 ears of corn { 42 ears of corn { 168

ears of corn { 189 ears of corn = 189 ears of corn remaining.

---

Additionally, two complete reasoning paths for MATH are also provided below. The overlapping section between these two reasoning paths, totaling **129** tokens, is highlighted in bold.

---

**Reasoning path 1 on MATH**

**$(x+1)^2+2(x+1)(3-x)+(3-x)^2$**

**$= (x^2+2x+1)+2(3-x)(x+1)+(9-6x+x^2)$**

**$= x^2+2x+1+2(3x+3-x^2-x)+(9-6x+x^2)$**

**$= x^2+2x+1+6x+6-2x^2-2x+9-6x+x^2$**

**$=** (x^2-2x^2+x^2)+(2x-2x+6x-6x)+(1+6+9)$

$= 0+0+16$

$= \boxed{16}$.

---

**Reasoning path 2 on MATH**

**$(x+1)^2+2(x+1)(3-x)+(3-x)^2$**

**$= (x^2+2x+1)+2(3-x)(x+1)+(9-6x+x^2)$**

**$= x^2+2x+1+2(3x+3-x^2-x)+(9-6x+x^2)$**

**$= x^2+2x+1+6x+6-2x^2-2x+9-6x+x^2$**

**$=** x^2-2x^2+x^2+2x-2x-6x+6+9-6$

$= \boxed{14}$.

# E  ABLATION STUDY

Table 8: Mean accuracy and standard deviation for Mistral 7B and Mistral 7B MetaMath on the GSM8K benchmark according to varying sizes of $K$ and $b$ when utilizing Step-by-Step Beam Search with the TVM. Three random trials are carried out.

| $K, b$ | FLOPs | Mistral 7B | Mistral 7B MetaMath |
|---|---|---|---|
| 20, 5 | $47.1 \times 10^{12}$ | 86.05 ±0.37 | 88.12 ±0.25 |
| 40, 10 | $94.3 \times 10^{12}$ | 87.69 ±0.22 | 88.70 ±0.16 |
| 80, 20 | $188.5 \times 10^{12}$ | 87.89 ±0.35 | 88.75 ±0.20 |
| 100, 25 | $235.6 \times 10^{12}$ | 87.92 ±0.13 | 88.80 ±0.07 |

**Beam size study.**  To investigate whether the accuracy of using the TVM improves with larger values of $K$ and $b$ in Step-by-Step Beam Search, we conduct experiments using the TVM with varying sizes of $K$ and $b$ for Mistral 7B and Mistral 7B MetaMath on the GSM8K benchmark. Table 8 shows that the accuracy of using the TVM on GSM8K increases as both $K$ and $b$ grow from 20 and 5 to 40 and 10. However, the accuracy of using the TVM remains relatively stable with the rise in $K$ and $b$ from 40 and 10 to 100 and 25, while the inference computation (i.e., FLOPs) increases by 2.5 times. Hence, 40 and 10 would be an appropriate choice for $K$ and $b$, considering the trade-off between FLOPs and the improved performance.

