# OpenReview forum: "Token-Supervised Value Models for Enhancing Mathematical Problem-Solving Capabilities of Large Language Models"
_ICLR.cc/2025/Conference — ICLR 2025 Poster_

### Official Review · Reviewer_fQ6h · 2024-11-01

**Soundness:** 3
**Presentation:** 3
**Contribution:** 4
**Rating:** 6
**Confidence:** 4

**Summary:**

The paper presents TVM -- token-supervised value models that can act as neural verifiers and provided tree-based LLM reasoning methods with a way to rank different parts of the search tree as more/less promising on reaching a solution in the domain of math problems. The authors state that existing SOTA like Outcome-based (ORMs) and Process-based (PRMs)  supervision models cannot rank paths effectively due to the way they are trained. ORMs only provide a reward if the answer is correct whereas PRMs provide the same reward to all steps. This causes issues in recall with PRMs since an entire step is marked as a failure even if only a few tokens are wrong.

To mitigate this drawback of PRMs, the authors introduce TVMs that utilizes different generated reasoning paths in the training set to provide different "weights" or rewards. For any substring of a correct solution, each token of that substring is allocated a weight based on the total substrings (or reasoning paths) that were generated. This can be thought of as a value function and the authors provide some soft theoretical analysis for it.

The authors then conduct an empirical evaluation that showcases the strengths of their approach.

**Strengths:**

S1. The paper is very well-written and organized although it seems a bit verbose with some aspects not really required.

S2. The presented idea is very convincing and intuitive. The idea is novel and weighing up of tokens (or a partial order sequence) has been successfully explored as an idea in many different domains.

S3. The baselines are good and the empirical setup overall is good.

**Weaknesses:**

W1. The paper was a bit verbose even if it was well-written. There are several aspects that could have been cut. For example, I do not understand why  Tables 4, 5 were included since it does not appear to be a major contribution of the paper. Table 3 already showcases all methods with different tree-search paradigms so I am not sure why this was necessary. Please comment. An analysis of training effort would have been better here.

W2. The empirical evaluation is a bit lacking. The gains seem marginal in most cases and std. deviations are missing from these plots. Please provide them so that a better assessment can be made.  Also, it seems that this approach can be more general. I am not sure why the authors have limited to only math tasks. A more general evaluation would have improved the paper's impact.

**Questions:**

Overall I liked this paper and I think it will be a good fit in the program. I hope the authors are able to resolve my concerns.
Please respond to my comments on the weaknesses.

---

> ### Author Response · Authors · 2024-11-23
> **Dear Reviewer fQ6h,**
>
> Dear Reviewer fQ6h,
>
> We really appreciate your constructive and insightful comments.
>
> -----------------------------
>
> **[Weakness 1. Reason why Tables 4 and 5 are included. An analysis of training effort would have been better.]**
>
> Thank you for bringing this to our attention. First of all, as elucidated in Sections 5.1 and 5.2, our original intention behind the inclusion of Table 4 in the original manuscript is to support the idea that tree-search-based inference methods can be more effective, or as effective as, the Best-of-N search while being much more efficient than Best-of-N search in terms of FLOPs and the execution time, especially when tree search strategies can be paired with an effective verifier like TVM. Along this line, to investigate whether the performance of step-level beam search with TVM improves with an increase in the number of $K$ and $b$, we have included Table 5 in the original manuscript.
>
> Reflecting on the reviewer’s insightful suggestion, we also agree that including an analysis of training efforts along with the analysis of inference efforts would improve the manuscript.
>
>
> Following the reviewer’s suggestion, using 8×NVIDIA A100-80GB GPUs, we estimated FLOPs and measured the execution time of sampling $N_{tr}$ reasoning paths to train TVMs on GSM8K and MATH for Mistral 7B MetaMath.
>
> <Table A. FLOPs and execution time of sampling $N_{tr}$ reasoning paths to train TVMs on the GSM8K ($N_{tr} = 100$) and MATH ($N_{tr} = 25$) benchmarks for Mistral 7B MetaMath>
>
> | | GSM8K FLOPs | GSM8K Time | MATH FLOPs | MATH Time |
> |:------------|:------:|:------:|:------:|:------:|
> | Sampling w/o vLLM | $130.4 \times 10^{13}$ | $8.2$ hours | $204.1 \times 10^{13}$ | $20.3$ hours |
> | Sampling w/ vLLM | $130.4 \times 10^{13}$ | $4.6$ hours | $204.1 \times 10^{13}$ | $5.7$ hours |
>
> We appreciate your valuable feedback. We have included Table A in Section 5.3 and have moved Table 5 in the original manuscript to Appendix E in the revised version.
>
> ---------------------------------------
>
> **[Weakness 2-1. Providing standard deviations in empirical evaluations would allow for a better assessment.]**
>
> Thank you for the helpful suggestion. We also agree that providing standard deviations is important for a better empirical evaluation. As the reviewer suggested, we first calculated the mean accuracy and standard deviation of TVM using step-level beam search and REBASE on GSM8K. These results were obtained from three random trials.
>
> <Table B. Mean accuracy and standard deviation of TVM on the GSM8K benchmark. Three random trials are conducted.>
>
> | Search Strategy | Mistral 7B | Mistral 7B MetaMath | Llama 3 8B | Llama 3 8B MetaMath |
> |:------------|:------:|:------:|:------:|:------:|
> | Step-level Beam Search | $87.69\small{\pm 0.22}$ | $88.70 \small{\pm 0.16}$ | $89.06 \small{\pm 0.07}$ | $90.35 \small{\pm 0.19}$ |
> | REBASE | $87.97 \small{\pm 0.16}$ | $89.21 \small{\pm 0.14}$ | $88.60 \small{\pm 0.09}$ | $89.84 \small{\pm 0.21}$ |
>
> We appreciate your valuable comment and have integrated the experimental results from Table B into Table 1 of the revised manuscript. In the final version, we will also include the mean accuracy and standard deviation of TVM on the MATH benchmark.
>
> -----------------------------------
>
> **[Weakness 2-2. It seems that the TVM approach can be applied to more general tasks.]**
>
> Theoretically, as the reviewer noted, the TVM approach is applicable to more general tasks. However, we first chose to focus on mathematical problem-solving tasks in this paper because, for math word problems, no automated tools are available to verify the exact correctness of candidate solutions when the ground truth answer is unavailable. In contrast, as highlighted in [1], tasks like code generation and neural theorem proving can benefit from automated tools, such as unit tests and proof assistants (e.g., Lean4), that can automatically determine the correctness of candidate solutions. Given this, we initially applied the TVM approach to math tasks and plan to extend it to more general tasks in future work.
>
> [1] Large Language Monkeys: Scaling Inference Compute with Repeated Sampling, arXiv:2407
>
> ---------------------------
>
> Once again, we sincerely appreciate your time and efforts in reviewing our paper. If you have any remaining issues or concerns, please do not hesitate to bring them to our attention.

---

> > ### Comment · Reviewer_fQ6h · 2024-11-25
> >
> > Thank you for your reponse. This has clarified all my concerns.
> >
> > Overall I am quite happy with this work. I think it is interesting and addresses an important problem.
> > I think that the paper's impact could be improved by considering other problems but I agree that it might be difficult to do that currently. If the paper gets rejected, I'd encourage the authors to improve and resubmit. All the best!

---

> > > ### Author Response · Authors · 2024-11-30
> > > **Dear Reviewer fQ6h,**
> > >
> > > Thank you so much for your consideration of our work. We are glad to hear that your concerns have been addressed and truly appreciate your support.

---

### Official Review · Reviewer_E5Q5 · 2024-11-03

**Soundness:** 3
**Presentation:** 3
**Contribution:** 3
**Rating:** 6
**Confidence:** 3

**Summary:**

This paper proposes token value models (TVMs), a new method for training verifiers used at test time to improve the problem-solving capabilities of LLMs. Existing training methods, outcome-supervised reward models (ORMs), and process-supervised reward models (PRMs) have drawbacks in that they can give only indirect supervision for intermediate steps. TVMs enable giving direct supervision over tokens by exploiting the probability that the token will read to a correct answer.  Experimental results show that TVM shows better predictive performance than ORMs and PRMs.

**Strengths:**

1. The proposed method is simple and easy to understand how it works.
2. Experiments are conducted with combinations of multiple benchmarks, LLMs, and search strategies. The experimental results support the proposed method's superiority.

**Weaknesses:**

Although the experimental results are interesting, the paper's presentation has some problems and thus does not explain the reasons why the TVM works well. Therefore, I think this paper needs a further revision.
  1. The paper says that TVM is superior to ORM and PRM since it uses token-level direct supervision (line 113). I'm not sure why we can say TVM is direct since ORMs, PRMs (without human annotations), and TVMs all rely on supervision signals for outcomes. No direct supervision on intermediate steps is given. Therefore, all these methods share the same problem described as the disadvantages of the ORMs (line 267).
  2. Moreover, the supervision of TVM (eq. (5)) based on conditional probability looks pretty similar to the reward for ORMs (eq. (3)). If we have $N_{tr}$ samples, the reward for a token $t_{n, k}$ in ORMs will be close to the success rate of the paths containing $t_{n, k}$. Therefore, the proposed supervision seems less novel.
  3. The paper says that one of its main contributions is that it is the first to disclose that PRMs produce a high false negative error. However, this is not true, as the explanation (line 306, Figure 2) assumes that PRMs do not use human annotation. The false-negative problem would not occur if we had human annotation on intermediate steps. Therefore, the statement is incorrect.

**Questions:**

1. (Figure 1) I need help understanding why ORM assigns low scores to correct steps and PRM assigns high scores to wrong intermediate steps based on this example. More convincing examples would help readers understand the superiority of the TVM.

2. I cannot understand the explanation of TVM's complexity (line 377). TVM is said to be a tree-search-based method (line 132), but the explanation here says that TVM computes eq.(6) from a set of fixed reasoning paths and does not need a tree search. I'm happy if the authors address this inconsistency.

3. Moreover, I think the roll-out process (line 374) is also beneficial for the TVM since we can estimate conditional probabilities eq.(6) more precisely for a token if we have more sample runs. Why does TVM avoid this?

---

> ### Author Response · Authors · 2024-11-23
> **Dear Reviewer E5Q5, [1]**
>
> Dear Reviewer E5Q5,
>
> We really appreciate your constructive and helpful comments.
>
> -------------------
>
> **[Weakness 1. Reason why the token-level supervision in TVM is direct.]**
>
> The reviewer mentioned that TVM uses token-level direct supervision (line 113). However, in line 113, we stated that we propose TVM to equip a verifier with a more direct ability to evaluate partial solutions. We did not claim that TVM uses token-level direct supervision or that the token-level supervision in TVM is direct (e.g. human-annotated).
>
> As the reviewer noted, ORMs, PRMs without human annotations, and TVMs all rely on outcome signals (i.e., 1 for a correct final answer and 0 otherwise) to construct supervision labels. Despite this shared dependency on supervision signals for outcomes, PRMs without human annotations outperform ORMs in Best-of-N search [1, 2] as they utilize step-wise distinct supervision compared to ORMs. Likewise, since TVMs leverage explicit token-level supervision with distinct correctness probability scores while outcome supervision in ORMs uses homogeneous labels determined by the correctness of a whole reasoning path, TVMs can more effectively assess whether a partial solution is progressing toward the correct answer during tree search at inference time. In this sense, TVMs do not suffer from the disadvantage of ORMs described in line 267, and we stated in line 113 that we propose TVMs to equip a verifier with a more direct ability to evaluate partial solutions.
>
> [1] Math-Shepherd: Verify and Reinforce LLMs Step-by-step without Human Annotations, ACL 2024
>
> [2] Multi-step Problem Solving Through a Verifier: An Empirical Analysis on Model-induced Process Supervision, EMNLP 2024 Findings
>
> ----------------------
>
> **[Weakness 2. The supervision of TVMs (Eq. 5) seems similar to the reward for ORMs (Eq. 3). Consequently, the reward for a token $t_{n,k}$ in ORMs will be close to the success rate of an intermediate reasoning path $\{t_{n,1}, \cdots, t_{n,k}\}$.]**
>
> While Eq. 3 in ORMs represents only the *cumulative reward*, Proposition 4.1 asserts that Eq. 5 in TVMs is equivalent to the *expected cumulative reward*. This key distinction highlights that our new token-level supervision scheme is practically different from the reward for ORMs. As seen in Table 1 of the paper, the TVM’s score for a token $t_{n,k}$ is closer to the actual success rate of an intermediate reasoning path ${t_{n,1}, \cdots, t_{n,k}}$, while the ORM’s reward for a token $t_{n,k}$ is less close to the actual success rate.
>
> -----------------------
>
> **[Weakness 3. The explanation in line 306 assumes that PRMs do not use human annotations, but the false-negative problem would occur for PRMs without human annotations.]**
>
> Thank you for bringing this to our attention. Since recent PRM studies do not rely on human annotations [1, 2, 3, 4] due to the high cost of human labeling, our paper also focuses primarily on **PRMs without human annotations**, as noted in line 292. Consequently, the term *'PRMs'* in line 306 actually refers to **PRMs without human annotations**. We appreciate your valuable comment and have updated the text to clarify this by replacing *'PRMs'* with *'PRMs without human annotations'* in lines 306-309. Moreover, to prevent further confusion, we have added the following clarification in line 310 of the revised manuscript: ``Hereafter, we refer to PRMs without human annotations simply as PRMs to keep the expression concise.’’
>
> [1] Math-Shepherd: Verify and Reinforce LLMs Step-by-step without Human Annotations, ACL 2024
>
> [2] Multi-step Problem Solving Through a Verifier: An Empirical Analysis on Model-induced Process Supervision, EMNLP 2024 Findings
>
> [3] AlphaMath Almost Zero: Process Supervision without Process, NeurIPS 2024
>
> [4] Improve Mathematical Reasoning in Language Models by Automated Process Supervision, arXiv:2406
>
> ----------------------
>
> **[Question 1. Clarification on Figure 1.]**
>
> Thank you for bringing this to our attention. First, since outcome supervision in ORMs labels all tokens in a reasoning path uniformly as either 0 or 1, ORMs can mispredict on challenging test problems, as shown in Figure 1. ORMs may reversely assign high scores close to 1 (e.g., 0.914 or 0.906) to incorrect intermediate steps and low scores close to 0 (e.g., 0.175 or 0.160) to correct ones. In contrast, PRMs without human annotations, which are trained with per-step correctness labels, avoid such reversed predictions. However, due to the fact that PRMs without human annotations are trained to determine an entire intermediate step as incorrect even if only the final few tokens are erroneous, they tend to under-value promising intermediate steps, assigning scores (e.g., 0.657 or 0.648) that are comparable to those of incorrect steps (e.g., 0.710 or 0.661). Consequently, both ORMs and PRMs perform suboptimally as verifiers for tree search strategies at inference time, as illustrated in Figure 1 and Sections 1 and 3.
>
> ---------------

---

> > ### Author Response · Authors · 2024-11-23
> > **Dear Reviewer E5Q5, [2]**
> >
> > --------------------------------
> >
> > **[Question 2. Clarification on the complexity of TVM.]**
> >
> > Thank you for bringing this to our attention. In line 132, we state that TVMs are suitable for tree-search-based **inference** methods. In contrast, computing Eq. 6 from the $N_{tr}$ generated reasoning paths pertains to **training** as written in line 361 and has a linear computational complexity as explained in line 377. In summary, line 132 focuses on inference, while line 377 is related to training.
> >
> > --------------------------------
> >
> > **[Question 3. Why don't TVMs use roll-outs?]**
> >
> > Theoretically, it is true that roll-outs can be conducted for TVMs to increase the fidelity of probability estimation. However, as delineated in line 376, this comes at the cost of increased (Quadratic) computational complexity. In practice, performing roll-outs is computationally burdensome. To streamline the annotation process, we use Eq. 6 to estimate token-level probabilities from the $N_{tr}$ sampled reasoning paths, which has a linear computational complexity as mentioned in line 377.
> >
> > -----------------------------
> >
> > Once again, we sincerely appreciate your time and efforts in reviewing our paper. If you have any remaining issues or concerns, please do not hesitate to bring them to our attention.

---

> > > ### Comment · Reviewer_E5Q5 · 2024-11-26
> > >
> > > Thank you for addressing my concerns. I think this paper is interesting, but I still cannot understand why the proposed method works well.  I have some additional questions. I'd be happy if the authors addressed them.
> > >
> > > ---
> > >
> > > **Additional Questions**
> > >
> > > **[ Weakness 1, 2]**
> > >
> > > > Likewise, since TVMs leverage explicit token-level supervision with distinct correctness probability scores while outcome supervision in ORMs uses homogeneous labels determined by the correctness of a whole reasoning path, TVMs can more effectively assess whether a partial solution is progressing toward the correct answer during tree search at inference time.
> > >
> > > > While Eq. 3 in ORMs represents only the cumulative reward, Proposition 4.1 asserts that Eq. 5 in TVMs is equivalent to the expected cumulative reward. This key distinction highlights that our new token-level supervision scheme is practically different from the reward for ORMs
> > >
> > > I'm still not sure why the authors say that the proposed method is better than the baseline methods. Maybe it is because I cannot understand the difference between TVM's *expected cumulative reward* and ORM's *cumulative reward*.  Could you please explain the mechanism of why the expected cumulative reward works better? In my understanding, both can set high scores for tokens leading to success, at least if we have infinitely many samples.
> > >
> > > **[Questions 3]**
> > > Related to question 3, I think it is also possible for PRM to achieve linear time complexity by not using rollouts. I'd like to know what happens if we use PRM without rollouts. If it leads to degradation, why does TVM not suffer from not using roll-outs?
> > >
> > > Answering these questions would help readers to understand why TVM works better than PRM and ORM.

---

> > > > ### Author Response · Authors · 2024-11-30
> > > > **Dear Reviewer E5Q5, [3]**
> > > >
> > > > We sincerely appreciate your invaluable feedback for enhancing the clarity of our paper.
> > > >
> > > > ------------------------
> > > >
> > > > **[Additional Question 1-1. Could you explain the mechanism of why the expected cumulative reward works better?]**
> > > >
> > > > Thank you for bringing this to our attention. As demonstrated in Proposition 4.1, since TVMs are trained using the expected cumulative reward, TVMs become a *value* model. Given that tree search is fundamentally intended to be guided by *value*, TVMs enable tree search algorithms to be value-guided, resulting in better performance during tree search. Consequently, utilizing the expected cumulative reward proves more effective when implementing tree search strategies.
> > > >
> > > > -------------------------
> > > >
> > > > **[Additional Question 1-2. Both the expected cumulative reward and the cumulative reward can set high scores for tokens leading to success if we have infinitely many samples.]**
> > > >
> > > > Thank you for the insightful comment. If we had an infinite number of samples and could train a verifier using full-batch (i.e., infinite batch size) gradient descent, then ORMs, PRMs without human annotations, and TVMs would all theoretically be equivalent. However, this scenario is purely hypothetical and practically infeasible for training verifiers. In real-world settings, where a finite number of samples is available, a verifier’s performance depends significantly on the supervision scheme employed. Consequently, the accuracy of ORMs, PRMs without human annotations, and TVMs differs empirically.
> > > >
> > > > --------------------------
> > > >
> > > > **[Additional Question 2. What happens if we use PRM without rollouts? If it leads to degradation, why does TVM not suffer from not using roll-outs?]**
> > > >
> > > > Thank you for the helpful suggestion.  In line with the reviewer’s suggestion, we compare the experimental results of *PRMs without human annotations both with and without roll-outs*.
> > > >
> > > > <Table A. Accuracy of Mistral 7B MetaMath on the GSM8K and MATH benchmarks under step-level beam search and REBASE, comparing PRMs without human annotations both with and without roll-outs.>
> > > >
> > > > | Benchmark | Search Strategy | Verifier | Mistral 7B MetaMath |
> > > > |:------------|:------------|:------------|:------:|
> > > > | GSM8K | Step-level Beam Search | PRMs without human annotations *with roll-outs* | $\mathbf{86.66}$ |
> > > > | | Step-level Beam Search | PRMs without human annotations *without roll-outs* | $86.13$ |
> > > > | GSM8K | REBASE | PRMs without human annotations *with roll-outs* | $\mathbf{86.28}$ |
> > > > | | REBASE | PRMs without human annotations *without roll-outs* | $85.67$ |
> > > > | MATH | Step-level Beam Search | PRMs without human annotations *with roll-outs* | $\mathbf{36.80}$ |
> > > > | | Step-level Beam Search | PRMs without human annotations *without roll-outs* | $36.00$ |
> > > > | MATH | REBASE | PRMs without human annotations *with roll-outs* | $\mathbf{37.60}$ |
> > > > | | REBASE | PRMs without human annotations *without roll-outs* | $37.00$ |
> > > >
> > > >
> > > > As the reviewer speculated, in PRMs without human annotations, not using roll-outs results in marginal accuracy degradation. In this context, we believe that TVMs without roll-outs offer a balanced approach between performance and computational complexity, given that the accuracy gain from using roll-outs would be marginal compared to the substantial increase in (quadratic) computational complexity.
> > > >
> > > > ---------------------------------
> > > >
> > > > Once again, thank you so much for taking the time and effort to participate in the discussion. If you have any further questions, please do not hesitate to reach out.

---

> > > > > ### Comment · Reviewer_E5Q5 · 2024-12-01
> > > > >
> > > > > Thank you for addressing my questions. I will increase my score.
> > > > >
> > > > > I understand that using expected cumulative reward is important because TVMs become a value model, which enables more effective tree searches.
> > > > >
> > > > > I understand that both PRMs and TVMs can run with and without rollouts, but it has been experimentally shown that TVMs without rollouts perform better than PRMs with rollouts. This is an interesting result. I recommend revising the description in lines 373-377  since it seems somewhat misleading: Roll-outs are not always necessary for PRMs, and TVMs may also benefit from roll-outs.

---

> > > > > > ### Author Response · Authors · 2024-12-01
> > > > > >
> > > > > > We are glad to hear that your concerns have been addressed. We sincerely appreciate your valuable comments, which have significantly contributed to the improvement of our paper. In response to the reviewer’s constructive feedback, we will revise the description in lines 373–377 in the final manuscript. Thank you so much for your consideration of our work.

---

### Official Review · Reviewer_6Dfx · 2024-11-04

**Soundness:** 3
**Presentation:** 3
**Contribution:** 3
**Rating:** 6
**Confidence:** 4

**Summary:**

The paper proposes Token-Supervised Value Models, to enhance the mathematical problem-solving capabilities of LLMs. Traditional methods lack effectiveness when applied to tree search algorithms, leading to premature pruning of promising steps. TVMs address this by assigning a probability to each token that reflects its likelihood of contributing to a correct answer, enabling direct evaluation of intermediate solutions. Experiments show that TVMs outperform ORMs and PRMs in accuracy, especially in tree-search-based strategies, by reducing false negatives and improving the assessment of solution paths​.

**Strengths:**

1. TVM is a novel approach. It assesses each token's contribution to the likelihood of reaching the correct answer, enabling more precise evaluations of intermediate solutions during tree search.

2. The authors address that TVMs lower false negative rates by distinguishing between promising and incorrect intermediate steps, enhancing recall without sacrificing precision.

3. TVMs show consistent improvements in accuracy across various benchmarks, including GSM8K and MATH, indicating their robustness and versatility.

**Weaknesses:**

1. TVMs require token-level supervision, which could add complexity to the training and fine-tuning process compared to traditional verifiers.

2. The effectiveness of TVMs relies on reasonably accurate token-level probability estimation based on samples of successful and failed trials. For difficult problems, the sampling process controlled by the LLM can be challenging, as most samples may be incorrect, potentially reducing the effectiveness of TVMs to that of a PRM.

3. Some sections need clarification; please see the questions below.

*Typo: In several places, there is an extra "p" added after "%", e.g., "14%p".

**Questions:**

1. How is $N_{tr}$ determined, given its direct impact on probability estimation? Is it chosen through parameter tuning, or is there additional insight guiding this choice?

2. A follow-up question: During fine-tuning, do these $N_{tr}$ trials often share some prefix tokens, and how to control them? For example, in Figure 4, the three paths share the first $a-1$ tokens, allowing those tokens to have the value 0.33. If they don’t share any tokens, it seems the problem would reduce to a PVM.


3. How to interpreted Figure 3(b)? The "Verifier’s Scores Histogram for Correct Sampled Solutions" shows that the weights are more concentrated for correct solutions, but what about the histogram on the right side?

---

> ### Author Response · Authors · 2024-11-23
> **Dear Reviewer 6Dfx,**
>
> Dear Reviewer 6Dfx,
>
> We greatly appreciate your constructive and helpful comments.
>
> ------------------------------------------------
>
> **[Weakness 1. Added complexity of token-level supervision to the training and fine-tuning process.]**
>
> To the best of our understanding, the reviewer’s concern is focused on the added *computational* complexity of token-level supervision during the training and fine-tuning of TVMs. Following [1], TVM is composed of an LLM backbone and an additional prediction head to predict token-wise probability scores. During the training phase, when the input sequence is fed forward, the prediction head generates predictions for each token, and the token-level supervision loss is computed. To evaluate the computational overhead introduced by this prediction head, we present a comparison of the parameter counts between the model backbone and the prediction head in Table A.
>
> <Table A. Comparison of the parameter counts between the model backbone and the prediction head for Mistral 7B and Llama 3 8B>
>
> | | Mistral 7B | Llama 3 8B |
> |:------------|:------:|:------:|
> | Backbone | $7241732096$ | $8030261248$ |
> | Head | $32002$ | $128258$ |
>
> We observe that the parameter count of the added linear prediction head is negligible compared to the verifier backbone. Thus, the added complexity is kept relatively small.
>
> [1] Training Verifiers to Solve Math Word Problems, arXiv:2110
>
> -----------------------------------------
>
> **[Weakness 2. For difficult problems, the effectiveness of TVMs would be reduced to that of PRMs.]**
>
> If an LLM is too small to solve difficult problems correctly even once among a large number of samples, nearly all samples would be incorrect, rendering the effectiveness of TVMs comparable to that of PRMs. However, even moderately sized LLMs (such as 7B-scale models) with some capability to solve problems that are difficult for humans make the sampling process less demanding, thus ensuring that most samples would not be incorrect. Notably, in practice, for Mistral 7B MetaMath and Llama 3 8B MetaMath, TVMs significantly outperform PRMs on the MATH benchmark—one of the most challenging benchmarks.
>
> -------------------------------------------
>
> **[Question 1. Choice of $N_{tr}$.]**
>
> As mentioned in Appendix C, we set $N_{tr}=100$ for GSM8K, following [1]. For MATH, we set $N_{tr}=25$ based on [2], where the number of solutions per training problem for MATH is set to one-fourth of that for GSM8K. Experimental results in Section 5 demonstrate that these $N_{tr}$ values are effective for training TVMs.
>
> [1] Training Verifiers to Solve Math Word Problems, arXiv:2110
>
> [2] Multi-step Problem Solving Through a Verifier: An Empirical Analysis on Model-induced Process Supervision, EMNLP 2024 Findings
>
> ------------------------------------------------------
>
> **[Question 2. Do $N_{tr}$ samples often share some prefix tokens? If they do not share any tokens, it seems that the effectiveness of TVMs would reduce to that of PRMs.]**
>
> As the reviewer astutely pointed out, if reasoning paths were sampled with a tiny $N_{tr}$, they might not share any tokens, leading the effectiveness of TVMs to reduce to that of PRMs. However, with a large enough $N_{tr}$ (e.g., $N_{tr} = 100$ for GSM8K as noted in the response to Question 1), we can obtain several reasoning paths that share prefix tokens. For instance, as exhibited in Figure 4, three reasoning paths share ```Terry took 8 bushels, Jerry took 3,```, which consists of 14 tokens in terms of token count. In addition, there are 17 reasoning paths that share ```Terry took 8```, which is composed of 5 tokens.
>
> ------------------------------------------
>
> **[Question 3. Interpretation of the right side of Figure 3 (b).]**
>
> The histogram on the right side of Figure 3 (b) illustrates the verifier score distribution for the intermediate steps of correct sampled solutions. Similar to the histogram on the left side of Figure 3 (b), it can be observed that PRM tends to under-value the scores of steps in correct solutions compared to TVM. Consequently, when employing PRMs with tree search strategies such as beam search, promising steps may be under-valued and subsequently pruned, resulting in lower performance.
>
> ----------------------------------------------
>
> Once again, we sincerely appreciate your time and efforts in reviewing our paper. If you have any remaining issues or concerns, please do not hesitate to bring them to our attention.

---

> > ### Author Response · Authors · 2024-12-01
> >
> > Dear Reviewer 6Dfx,
> >
> > We hope this message finds you well.
> >
> > We sincerely appreciate the time and effort you have dedicated to reviewing our paper. With less than 48 hours remaining before the deadline for reviewers to post messages to the authors, we kindly remind you of our response to your constructive and helpful comments. If you have any remaining questions or concerns, please feel free to reach out.
> >
> > Best regards,
> >
> > Authors

---

### Official Review · Reviewer_S2DY · 2024-11-07

**Soundness:** 3
**Presentation:** 3
**Contribution:** 2
**Rating:** 6
**Confidence:** 4

**Summary:**

This paper proposes token-supervised value models (TVMs) as a superior way to verify whether tree-search based math reasoning models are on the right track or not. The authors compare TVMs to output-supervised reward models (ORMs) and process-supervised reward models (PRMs). They argue that both of these existing verification methods, having been designed for evaluation of full trajectories, are not suitable for verifying partial trajectories, especially at the token level. The conduct experiments with two math datasets, showing that the use of TVMs improves performance. Additional experiments are also provided.

**Strengths:**

* The authors make a clear argument about what current verifiers (ORMs and PRMs) lack, and why TVMs should be able to help. Limitations of ORMs and PRMs are justified with some empirical support. Logically thinking, the progression from ORMs to PRMs to TVMs seems natural.

* Figures and examples (although small and hard to read) are useful.

* The method is clearly described, along with a formulation of prior (ORM and PRM) methods.

* Experiments show a notable improvement when using TVMs, especially for step-level beam search and for the MATH benchmark.

* Additional experiments around FLOPs used and execution time support the idea that tree-based search methods are more efficient, especially if one can pair them with effective verifiers.

**Weaknesses:**

* One obvious disadvantage of TVM compared to PRM is that during *training*, the same N_{tr} sampled reasoning paths per training problem are expected to provide a lot more supervision in TVM. Specifically, suppose a typical reasoning path has 5 steps and 50 tokens. Then, in PRMs, it is easy to imagine several reasoning paths sharing similar partial paths  steps, leading to a good signal for supervising the 5 steps. However, with ~50 token long paths, the chances of partial sampled paths overlapping would decreases quickly and would be very slim after the first 15-20 tokens. So I'm not sure what kind of token-supervision signal can one actually get for the latter parts of the reasoning paths -- unless paths are sampled differently or N_{tr} is higher. The paper doesn't seem to address / discuss this.

* The gains, when using TVM, are rather slim in step-level beam search for the GSM8k dataset, only about 1%. This is not very different from the gains when using best-of-N search. If fact, for Mistral, the gains from using TVM are slightly higher for best-of-N compared to step-level beam search. This goes against the motivational argument presented earlier, namely that ORMs and PRMs are suitable for best-of-N search but not tree search, and it's the latter than needs another verification method.

* In general, TVM doesn't seem to help that much on GSM8k.

* I don't understand what Proposition 4.1 is trying to say. Why is it not "trivially true" if the reward function is 0 at intermediate tokens and 1 at the output token iff is it the correct output? Is this proposition related to TVMs or also equally applicable to ORMs and PRMs (which seems to be the case)? I am not following the significance of this formal proposition.

**Questions:**

Please see the weaknesses section above. I am happy to consider increasing my score if I can get more clarity.

====
REVIEW RATING UPDATED AFTER AUTHOR RESPONSE
====

---

> ### Author Response · Authors · 2024-11-23
> **Dear Reviewer S2DY, [1]**
>
> Dear Reviewer S2DY,
>
> We greatly appreciate your constructive and insightful comments.
>
> -------------------------------------------------------------
>
> **[Weakness 1-1. Discussion about the token-supervision signal for the latter part of reasoning paths.]**
>
> As the reviewer astutely pointed out, if reasoning paths were sampled with a small $N_{tr}$, the likelihood of reasoning paths overlapping after the first couple of dozen (e.g., 15–20) tokens could be low. In such a case, the token-supervision signal for the latter part of each reasoning path would be labeled uniformly as either 1 or 0, depending on whether the final answer is correct.
>
> However, with a large enough $N_{tr}$ (e.g., $N_{tr} = 100$ for GSM8K as in [1]), we can obtain reasoning paths that overlap significantly beyond the first 15-20 tokens, up to the later steps of reasoning. As demonstrated in Figure 2 of the paper, reasoning paths can overlap up to the middle of the fourth step - which consists of the first **93** tokens in terms of token count. Below, we provide the complete version of the two overlapping reasoning paths that were illustrated in Figure 2. Bold text indicates the overlapped part between two reasoning paths in Figure 2, which is **93** tokens in total. Additionally, two complete reasoning paths for MATH are also provided below. The overlapping section between these two reasoning paths, totaling **129** tokens, is highlighted in bold.
>
>
> [1] Training Verifiers to Solve Math Word Problems, arXiv:2110
>
> ```Reasoning path 1 on GSM8K in Figure 2 (Correct)```
>
> **Terry took 8 bushels * 14 ears of corn per bushel = 112 ears of corn.**
>
> **Jerry took 3 bushels * 14 ears of corn per bushel = 42 ears of corn.**
>
> **Linda took 12 bushels * 14 ears of corn per bushel = 168 ears of corn.**
>
> **Stacy took 21 ears of corn**.
>
> Altogether, Bob's friends took 112 + 42 + 168 + 21 = 343 ears of corn.
>
> Thus, Bob is left with 50 bushels * 14 ears of corn per bushel - 343 ears of corn = 357 ears of corn.
>
>
> ```Reasoning path 2 on GSM8K in Figure 2 (Wrong)```
>
> **Terry took 8 bushels * 14 ears of corn per bushel = 112 ears of corn.**
>
> **Jerry took 3 bushels * 14 ears of corn per bushel = 42 ears of corn.**
>
> **Linda took 12 bushels * 14 ears of corn per bushel = 168 ears of corn.**
>
> **Stacy took 21 ears of corn** + 168 ears of corn = 189 ears of corn.
>
> Bob had 50 bushels * 14 ears of corn per bushel = 700 ears of corn to start with.
>
> So, Bob has 700 ears of corn – 112 ears of corn – 42 ears of corn – 168 ears of corn – 189 ears of corn = 189 ears of corn remaining.
>
>
> ```Reasoning path 1 on MATH (Correct)```
>
> $\mathbf{(x+1)\^2+2(x+1)(3-x)+(3-x)^2}$
>
> $\mathbf{= (x^2+2x+1)+2(3-x)(x+1)+(9-6x+x^2)}$
>
> $\mathbf{= x^2+2x+1+2(3x+3-x^2-x)+(9-6x+x^2)}$
>
> $\mathbf{= x^2+2x+1+6x+6-2x^2-2x+9-6x+x^2}$
>
> $\mathbf{=}$ $(x^2-2x^2+x^2)+(2x-2x+6x-6x)+(1+6+9)$
>
> $= 0+0+16$
>
> $= \boxed{16}$.
>
>
> ```Reasoning path 2 on MATH (Wrong)```
>
> $\mathbf{(x+1)^2+2(x+1)(3-x)+(3-x)^2}$
>
> $\mathbf{= (x^2+2x+1)+2(3-x)(x+1)+(9-6x+x^2)}$
>
> $\mathbf{= x^2+2x+1+2(3x+3-x^2-x)+(9-6x+x^2)}$
>
> $\mathbf{= x^2+2x+1+6x+6-2x^2-2x+9-6x+x^2}$
>
> $\mathbf{=}$ $x^2-2x^2+x^2+2x-2x-6x+6+9-6$
>
> $= \boxed{14}$
>
> We appreciate your valuable feedback and have revised our manuscript to include this point and the above full reasoning paths in Appendix D.
>
>
> --------------------------------------------
>
> **[Weakness 1-2. The same $N_{tr}$ sampled reasoning paths per training problem are expected to provide more supervision in TVM, which seems a disadvantage.]**
>
> The reviewer mentioned that the same $N_{tr}$ sampled reasoning paths per training problem are expected to provide more supervision in TVM, but this only pertains to the case when $N_{tr}$ is small, as discussed in Weakness 1-1. Rather, it is important to emphasize that when using a large enough $N_{tr}$, token-level supervision signals can be effectively obtained (as also highlighted in Weakness 1-1). In this regime, TVM benefits from more detailed token-level supervision, even for the latter parts of reasoning paths. In our experiments, setting a commonly utilized $N_{tr}$ as in [1] was capable of providing this detailed supervision, which has been empirically shown to enable TVMs to outperform existing verifiers (ORMs and PRMs) when using tree search methods.
>
> [1] Training Verifiers to Solve Math Word Problems, arXiv:2110
>
> -------------------------------------------------------------

---

> > ### Author Response · Authors · 2024-11-23
> > **Dear Reviewer S2DY, [2]**
> >
> > ---------------------------------------
> >
> > **[Weakness 2. For GSM8K, the accuracy gain of tree search when using TVM seems not much different from that of best-of-N search, and for Mistral 7B, it is slightly smaller, which appears to go against the paper’s motivational argument.]**
> >
> > The reviewer observed that for GSM8K, the accuracy gain of tree search using TVM appears similar to that of best-of-N search and that for Mistral 7B on GSM8K (in Table 2 of the paper), tree search using TVM shows a smaller accuracy gain than best-of-N search, seemingly challenging the paper's motivational argument. However, it is important to note that PRM was originally proposed as a verifier to demonstrate stronger performance compared to ORM in best-of-N search. Therefore, the accuracy gain of best-of-N search using TVM should be considered between PRM and TVM.
> >
> > For Mistral 7B MetaMath on GSM8K in Table 2 of the paper, where TVM can be compared with PRM in best-of-N search, the accuracy gain of best-of-N search when using TVM instead of PRM is less than 0.5%, while the gain of step-level beam search is around 1.0%. This trend is more distinctively observed for the MATH dataset, where the accuracy gain of best-of-N search when using TVM is almost negligible while the gain of step-level beam search is around **2.4%**. Based on this, we can conclude that the accuracy gain of step-level beam search when using TVM is indeed larger than that of best-of-N search when using TVM, which supports our motivational argument.
> >
> > --------------------------------------------
> >
> > **[Weakness 3. TVM does not seem to help that much on GSM8K (only about 1% accuracy gain).]**
> >
> > As pointed out by the reviewer, TVM is less effective on GSM8K, while being more effective on MATH. This may be due to the fact that the average number of steps required for GSM8K is $4.5$, whereas for MATH, it is $11.0$, as referenced in [1]. Since GSM8K requires less than half the average steps of MATH, and verifiers can only intervene after each step, the number of verifier interventions on GSM8K is consequently lower. As a result, TVM appears less beneficial for GSM8K. However, it is worth noting that TVM proves significantly helpful on MATH, which is a more challenging benchmark with more room for improvement.
> >
> > [1] Multi-step Problem Solving Through a Verifier: An Empirical Analysis on Model-induced Process Supervision, EMNLP 2024 Findings
> >
> > --------------------------------------------------------
> >
> > **[Weakness 4. Significance of Proposition 4.1.]**
> >
> >
> > Thank you for bringing this to our attention. First, Proposition 4.1 pertains only to TVMs and does not apply to existing reward models (ORMs and PRMs), because Proposition 4.1 ensures that TVM is a *value* model. Given that tree search is fundamentally intended to be guided by *value* rather than *reward*, we believe that Proposition 4.1 is important, as Proposition 4.1 guarantees that TVMs allow tree search algorithms to be value-guided.
> >
> > As the reviewer noted, the reward function generally assigns a reward of 0 to intermediate tokens and 1 to the output token if it is correct, and vice versa. However, for an incorrect output, the reward function can be designed in two ways in reinforcement learning: (i) 0 for intermediate tokens and 0 for the output token, or (ii) 0 for intermediate tokens and -1 for the output token. For our approach, we adopt the former design and define the reward function as in Eq. (1) in the paper.
> >
> > We appreciate your valuable feedback and have clarified this point in **Section 4.2** of the revised manuscript.
> >
> > ------------------------------------------------
> >
> > Once again, we sincerely appreciate your time and efforts in reviewing our paper. If you have any remaining issues or concerns, please do not hesitate to bring them to our attention.

---

> ### Author Response · Authors · 2024-12-01
>
> Dear Reviewer S2DY,
>
> We hope this message finds you well.
>
> We sincerely appreciate the time and effort you have dedicated to reviewing our paper. With less than 48 hours remaining before the deadline for reviewers to post messages to the authors, we kindly remind you of our responses to your constructive and insightful comments. If you have any remaining questions or concerns, please feel free to reach out.
>
> Best regards,
>
> Authors

---

> > ### Comment · Reviewer_S2DY · 2024-12-03
> > **Re: Official Comment by Authors**
> >
> > Thank you for the explanations and examples.  This helps me see that there is indeed a reasonable possibility of token-level overlap in ~100 sampled paths even during the later parts of the paths, at least on the GSM8K and MATH benchmarks (that this actually happens in practice still surprises me, but I see it in your examples).  I also see the value of proposition 4.1. I don't quite follow your argument for weakness 2, but nevertheless will raise my score to reflect my assessment after improved understanding.  Thank you and good luck!

---

> > > ### Author Response · Authors · 2024-12-04
> > >
> > > Thank you very much for your consideration of our work. We also appreciate you pointing out our response to Weakness 2. In summary, our intent in addressing Weakness 2 was to emphasize that, when considering the accuracy gain from using TVM in best-of-N search as the accuracy difference between PRM and TVM in best-of-N search, the gain is actually smaller than that in step-level beam search.
> > >
> > > Once again, we sincerely thank you for your invaluable feedback, which has significantly contributed to improving the clarity of our paper. In response to the reviewer’s constructive comment on our handling of Weakness 2, we will ensure that this point is further clarified in the final manuscript.

---

### Meta-Review · Area_Chair_7635 · 2024-12-24

**Metareview:**

This work introduces token-supervised value models (TVMs), which provide token-level supervision using verifiers. The verifiers assign a score to each token, indicating the probability of reaching a correct final answer. The authors show that TVMs achieve lower false negative errors than process-supervised reward models without human annotations, and provide interesting theoretical insights that their verifiers are equivalent to value functions, which can be used to guide tree search.

The reviewers suggested that the proposed idea is novel and convincing, the experiments show good performance, and the paper is well-written in general with clear descriptions. However, some reviewers also noted issues about training complexity, low performance gain on some tasks and verbosity in writing. The authors actively engaged with reviewers during the rebuttal and discussion phase.

Overall I think the strengths outweigh the weaknesses. All reviewers leaned towards acceptance and I concur.

**Additional Comments On Reviewer Discussion:**

The reviewers suggested that the proposed idea is novel and convincing, the experiments show good performance, and the paper is well-written in general with clear descriptions. However, some reviewers also noted issues about training complexity, low performance gain on some tasks and verbosity in writing. The authors actively engaged with reviewers during the rebuttal and discussion phase.

Overall I think the strengths outweigh the weaknesses. All reviewers leaned towards acceptance and I concur.

---

### Decision · Program_Chairs · 2025-01-22

Accept (Poster)